# A high-fat and fructose diet in dogs mirrors insulin resistance and β-cell dysfunction characteristic of impaired glucose tolerance in humans

**Justin M. Gregory**[1]*, **Guillaume Kraft**[2], **Chiara Dalla Man**[3], **James C. Slaughter**[4], **Melanie F. Scott**[2], **Jon R. Hastings**[2], **Dale S. Edgerton**[2], **Mary C. Moore**[2], **Alan D. Cherrington**[2]

1 Ian Burr Division of Pediatric Endocrinology and Diabetes, Vanderbilt University School of Medicine, Nashville, TN, United States of America, 2 Department of Molecular Physiology and Biophysics, Vanderbilt University School of Medicine, Nashville, TN United States of America, 3 Department of Information Engineering, University of Padova, Padova, Italy, 4 Department of Biostatistics, Vanderbilt University Medical Center, Nashville, TN, United States of America

* justin.m.gregory.1@vumc.org

**Data Availability Statement:** All relevant data are within the paper and its Supporting Information files.

## Abstract

This study examined the impact of a hypercaloric high-fat high-fructose diet (HFFD) in dogs as a potential model for human impaired glucose tolerance (IGT) and type 2 diabetes mellitus (T2DM). The HFFD not only led to weight gain but also triggered metabolic alterations akin to the precursors of human T2DM, notably insulin resistance and β-cell dysfunction. Following the HFFD intervention, the dogs exhibited a 50% decrease in insulin sensitivity within the first four weeks, paralleling observations in the progression from normal to IGT in humans. Calculations of the insulinogenic index using both insulin and C-peptide measurements during oral glucose tolerance tests revealed a significant and sustained decrease in early-phase insulin release, with partial compensation in the later phase, predominantly stemming from reduced hepatic insulin clearance. In addition, the Disposition Index, representing the β-cell's capacity to compensate for diminished insulin sensitivity, fell dramatically. These results confirm that a HFFD can instigate metabolic changes in dogs akin to the early stages of progression to T2DM in humans. The study underscores the potential of using dogs subjected to a HFFD as a model organism for studying human IGT and T2DM.

## Introduction

Impaired glucose tolerance (IGT) and type 2 diabetes mellitus (T2DM) are multifaceted metabolic disorders typified by insulin resistance and hyperglycemia, which affect an estimated 537 million adults, equal to 1-in-10 adults worldwide [1]. These conditions often evolve in tandem with obesity, underscoring the need to elucidate the metabolic shifts induced by diet-related obesity to devise effective therapeutic strategies [2]. However, the translational potential of these strategies is often constrained by the inability of preclinical animal models to accurately reproduce human disease phenotypes [3].

**Funding:** J.M.G. has received career development awards and support from the National Institute of Diabetes and Digestive and Kidney Diseases of the National Institutes of Health (K23DK123392), Vanderbilt Diabetes Research and Training Center (DK020593), a JDRF Career Development Award (5-ECR-2020-950-A-N). A.D.C. has received research funding from R01DK018243. These studies were drawn from research supported by Fractyl Health and Metavention. The funders had no role in the study design, data collection and analysis, decision to publish, or preparation of the manuscript for these experiments. The contents are solely the responsibility of the authors and do not necessarily represent official views of the National Institute of Diabetes and Digestive and Kidney Diseases, the National Institutes of Health, or the JDRF.

**Competing interests:** Justin Gregory has served as an advisory board member for Sanofi, Eli Lilly, Medtronic, Dompe, vTv Therapeutics, and Mannkind Corporation and in data and safety monitoring roles for vTv Therapeutics and Medtronic. Alan Cherrington is a scientific advisor, consultant, and holds stock in Fractyl Laboratories. He also holds stock in Metavention. The other authors have declared that no competing interests exist. This does not alter our adherence to PLOS ONE policies on sharing data and materials.

Murine models, such as the leptin receptor deficient db/db mouse, are a popular and well-established choice for metabolic syndrome and T2DM research given their genetic tractability and the ability to perform high-throughput, cost-effective studies [4,5]. However, it is important to note that the obesity phenotype in these models, predominantly driven by genetic mutations, differs from the multifactorial nature of obesity typically associated with human T2DM. While these models can replicate aspects such as hyperglycemia and insulin resistance, the distinct physiological and genetic mechanisms underlying glucose metabolism in humans and mice pose challenges to their translational applicability [6]. For instance, unlike humans, mice display only a transient insulin response during oral glucose tolerance tests (OGTT) and rapidly absorb and clear enterically derived glucose from the blood [7]. Further, whereas humans exhibit a robust and sustained suppression of endogenous glucose production in response to an oral glucose load, mice have little if any suppression of endogenous glucose production during an OGTT [7]. Additionally, the monogenic and inbred nature of most murine models contrasts with the polygenic nature of human prediabetes and T2DM [8].

In contrast, dogs offer a promising model for studying human metabolic disorders. With a level of genetic diversity similar to that of humans [9], canine models can provide a broader understanding of how potential therapies might perform in a genetically diverse human population. Dogs and humans share substantial similarities in insulin structure, glucose dynamics, pancreas and islet structures, and β-cell replication capacities [10]. Importantly, dogs consuming a high-fat, high-fructose diet (HFFD) exhibit overweight conditions, insulin resistance, and IGT—mirroring the metabolic responses seen in humans on a similar diet [11]. This diet also leads to an increase in visceral, subcutaneous, and total adipose tissue mass, paralleling observations in human metabolic syndrome and T2DM [10,12]. Consequently, the HFFD canine model has the potential to accurately represent critical aspects of human IGT and T2DM, providing translatable insights into disease pathophysiology and therapeutic strategies.

Building on these compelling parallels, our lab has employed the HFFD canine model for preclinical testing of diabetes therapies and devices over the past fifteen years. In this study, we delve deeper into the metabolic response of dogs on the HFFD, utilizing serial OGTTs to examine the potential of the HFFD canine model to emulate key characteristics of human IGT and T2DM.

We hypothesized that the HFFD intervention in dogs would accurately mimic the key attributes of human IGT, specifically assessing the alterations in insulin sensitivity in fasted and absorptive states, and the ability to raise insulin in response to an enteric glucose load. Through this investigation, we seek to enhance our understanding of the pathophysiology of these diseases, to inform the development of more effective therapeutic strategies, and to provide a model others can use.

## Materials and methods

### Animals and experimental timeline

We examined the metabolic impact of a high-fat high-fructose diet (HFFD) in 32 dogs prior to preclinical therapeutic and device testing. All protocols received approval from the Vanderbilt Institutional Animal Care and Use Committee. Dogs were housed in an Association for Assessment and Accreditation of Laboratory Animal Care-accredited animal facility and were maintained on a 12:12 light:dark cycle, with temperatures between 64–84°F (17.8–28.9°C) and a relative humidity of 30–70%. Adult mongrel hounds (Marshall BioResources, North Rose, NY) of both sexes underwent a two-week acclimation period that included a daily diet of canned food (400 g) and chow (550 g; LabDiet Laboratory Canine Diet 5006, PMI Nutrition International, LLC, Brentwood, MO). Filtered municipal water via an

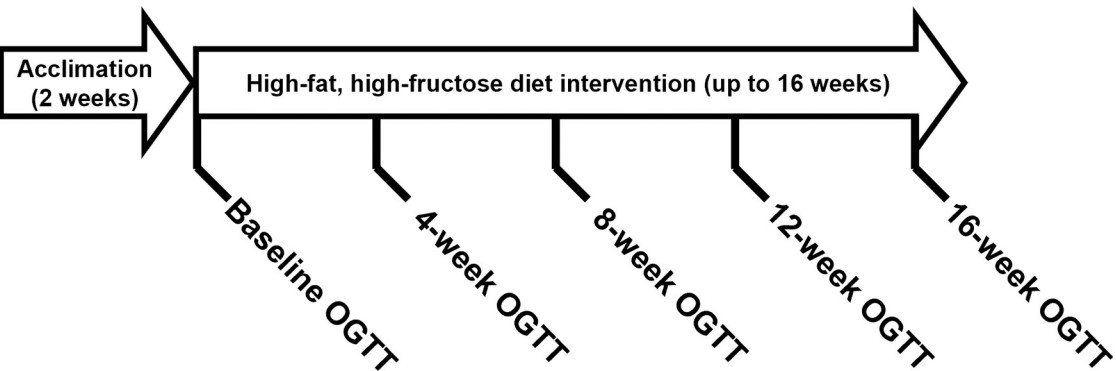

**Fig 1. Experimental timeline.**

automated watering system was always available. Animals underwent routine vaccination (DA2PP, Leptospira, rabies, and Bordetella) and parasite treatment (amprolium, sulfadimethoxine, pyrantel pamoate, and ivermectin) at the vendor prior to shipping. All dogs received an intake exam by a veterinarian within 48 hours of arrival to ensure overall health prior to study enrollment.

Following the acclimation period, we performed an oral glucose tolerance test (OGTT) on each dog to establish baseline metabolic parameters, including glucose tolerance and insulin sensitivity. Subsequently, the dogs were transitioned to a hypercaloric HFFD, composed of 20% protein, 53% fat, and 27% carbohydrate, with fructose accounting for 17% of the total energy (PMI Nutrition TestDiet, St. Louis, MO). OGTTs were repeated at varying four-week intervals following the commencement of the HFFD intervention (Fig 1). In this investigation, we retrospectively analyzed existing data, where all dogs received the HFFD intervention. Because of the retrospective nature of the analysis, a control diet was not included in the study design and no blinding, a priori sample size calculation, or randomization procedure was employed.

## Experimental design

We evaluated dogs that had undergone an 18-hour overnight fast and met health criteria including leukocyte count <18,000/mm$^3$, hematocrit >35%, good appetite, normal stooling, and sound physical appearance. No dogs required exclusion based on these criteria. On the morning of experiment, an angiocatheter was placed within a peripheral leg vein and advanced into a deep venous location, allowing for serial sampling of plasma glucose, insulin, and C-peptide. The dogs then rested in a Pavlov harness for a 60-minute acclimatation period. Each OGTT started with a 10-minute basal sampling period, followed by an oral administration of 0.9 g glucose/kg baseline weight (SolCarb, Medica Nutrition, Englewood, NJ). Over the subsequent 180 minutes, we collected serial plasma samples during nutrient absorption.

To ensure daily metabolic consistency and to assess dietary intake, canines were housed singly and provided with 4–6 hours daily socialization with other dogs. During housing and experiments dogs were provided with manipulanda (e.g., rubber and plastic chew toys, plastic balls), daily positive human interaction, and sensory enrichment (music or nature sounds). Once the series of OGTTs were completed, dogs subsequently underwent testing of differing diabetes therapies and devices, which are not included in this analysis. After completing the preclinical testing, dogs were euthanized using sodium pentobarbital.

## Analytical procedures

Plasma venous glucose, insulin, and C-peptide concentrations were measured as described elsewhere [13].

## Calculations

We used trapezoidal approximation to calculate the change in area under the curve (ΔAUC) above baseline for plasma venous glucose, insulin, and C-peptide concentrations. We then determined the following parameters to quantify the metabolic response to the HFFD intervention:

- The <u>Homeostatic Model Assessment of Insulin Resistance (HOMA-IR)</u> is a mathematical model used to estimate insulin resistance based on fasting glucose and fasting insulin levels [14]. We quantified HOMA-IR as basal insulin (μU/mL) × basal glucose (mg/dL)/405.

- The <u>Insulinogenic Index</u> is a measure of beta-cell function and insulin secretion in response to a glucose challenge, calculated as $\Delta AUC_{insulin} / \Delta AUC_{glucose}$ (in pmol insulin/mmol glucose) and as $\Delta AUC_{C\text{-}peptide} / \Delta AUC_{glucose}$ (in pmol C-peptide/mmol glucose) [15]. We quantified this parameter for the first 30 and 120 minutes following the enteric glucose load.

- The <u>Oral Glucose Minimal Model</u> is a mathematical model that quantifies insulin sensitivity ($S_I$), β-cell responsivity (Φ), and the disposition index (DI). Using custom scripts written in MATLAB, we estimated the following indices:

  ○ <u>Insulin sensitivity</u> ($S_I$, in $10^{-4} \cdot$ dL/kg/min per μU/mL), measuring the effect of insulin to enhance glucose disposal and inhibit hepatic glucose production [16], was determined by fitting the Oral Glucose Minimal Model to the glucose and insulin data.

  ○ <u>β-cell responsivity</u> (Φ, in $10^{-9} \cdot min^{-1}$), measuring the ability of glucose to stimulate pancreatic insulin secretion, was estimated by fitting the Oral C-peptide Minimal Model [17] to C-peptide and glucose data. The model relies on a two-compartment model of C-peptide kinetics based on the formulas proposed by Van Cauter et al. [18], which characterizes the distribution and clearance of C-peptide in the body and is parameterized using age and BMI values. In the absence of these measures in dogs, we assumed an age of 15 years and a BMI of 20 for all dogs, corresponding to the parameters of lean, young humans.

  ○ <u>Disposition Index</u> (DI in $10^{-13} \cdot$ dL/kg min$^{-2}$ per mU/mL) is the product of $S_I$ and Φ and represents the ability of the β-cells to compensate for changes in insulin sensitivity.

## Statistical analysis

We estimated the effects of the HFFD on the metabolic parameters over time using a linear mixed effects model (LMM). For each outcome (weight, insulinogenic index, HOMA-IR, and oral glucose minimal model parameters), we considered the week of the study as a fixed effect and the dog-specific deviations from the overall intercept as a random effect. We assumed a compound symmetric covariance structure for repeated measures on the same dog over time. Each week was modeled using indicator variables (4 degrees of freedom) and tests were relative to baseline (week 0). All statistical analyses were performed using IBM SPSS Statistics, version 28.0 (IBM Corp., Armonk, NY, USA). Data are summarized as medians (interquartile range) unless otherwise noted. Study data and the ARRIVE 2.0 checklist are included in a supplemental dataset.

## Results

### Changes in body weight

At the beginning of the study, the dogs had a median baseline weight of 23.0 (22.1–24.0) kg. The HFFD intervention triggered a marked shift in body weights over a period of 16 weeks (Fig 2). The dogs gained weight in proportion to the duration of the HFFD intervention, with the median weight peaking at 27.4 (25.1–28.7) kg at week 16 (p < 0.001 vs. week 0).

### Alterations in plasma venous glucose, insulin, and C-peptide concentrations during OGTTs

The implementation of the HFFD intervention triggered substantial changes in plasma venous glucose, insulin, and C-peptide concentrations. Plasma venous glucose concentrations during the OGTT doubled in the initial four weeks of the HFFD (p < 0.001), peaking at week 8. They then moderately decreased by 12 and 16 weeks of diet, remaining 65% higher than baseline at week 16 (p = 0.007) (Fig 3A and 3D). However, fasting plasma glucose concentrations displayed minimal variation, from 110.0 (106.0–115.3) mg/dL (6.1 mmol/L, IQR 5.9–6.4 mmol/L) at baseline to 107.0 (105.5–109.8) mg/dL (5.9 mmol/L IQR 5.9–6.1 mmol/L) at Week 16.

Plasma insulin concentrations during the OGTT underwent a significant increase of nearly one-third between the baseline and Week 4 OGTT (p < 0.001) (Fig 3B and 3E). While a decline in insulin ΔAUC was observed at weeks 8 and 12 (p = 0.041 and p = 0.036 vs baseline, respectively), the Week 16 OGTT demonstrated a renewed increase (p = 0.004 vs. baseline). Despite these changes, fasting plasma insulin concentrations remained relatively consistent throughout the intervention, registering 5.2 (4.0–6.8) μU/mL (31.2 pmol/L, IQR 24.0–40.8 pmol/L) at baseline and 6.7 (5.0–7.7) μU/mL (40.2 pmol/L, IQR 20.0–46.2 pmol/L) at Week 16.

Venous plasma C-peptide concentrations displayed a more moderate increase during the HFFD intervention compared to insulin (Fig 3C and 3F). An approximately 20% rise in C-

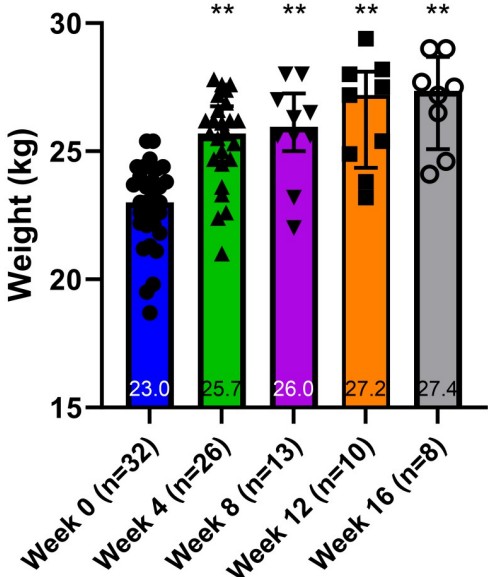

**Fig 2. Aggregate weight measurements during high-fat, high-fructose intervention.** Graph depicts medians (interquartile range). ** Denotes p < 0.001 vs. Week 0.

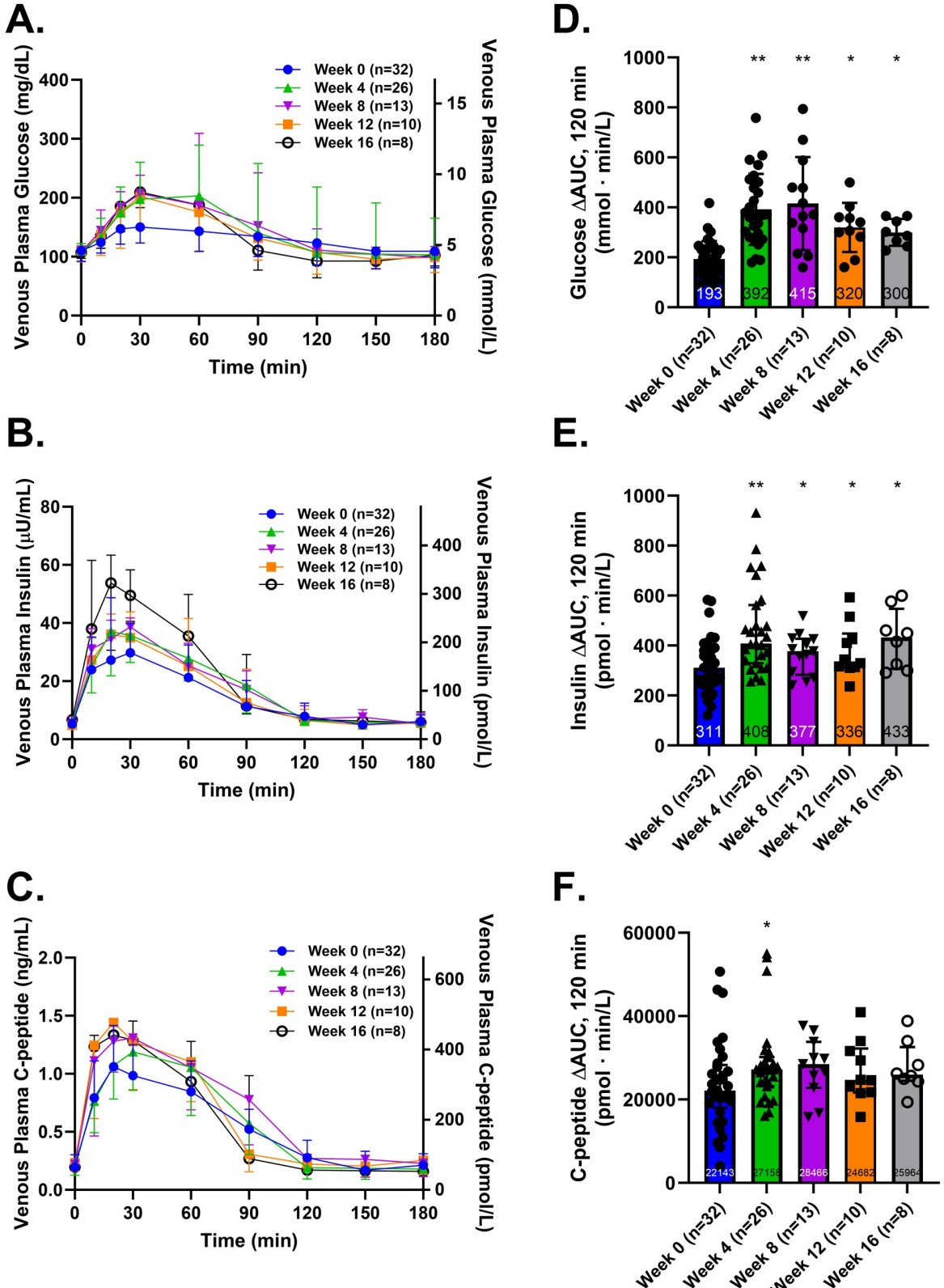

**Fig 3. Canine OGTT experiments.** Dogs were fed a high fat, high fructose diet (HFFD) to emulate the Western diet and were studied longitudinally. Venous plasma glucose (A), insulin (B), and C-peptide (C) were serially measured and the change in the area of curve above baseline was quantified (depicted in D-F, respectively). Graphs depict medians (interquartile range). * denotes $p < 0.05$ and ** denotes $p < 0.001$ vs Week 0.

peptide AUC occurred between baseline and week 4 OGTT ($p = 0.024$), which subsequently remained stable until the end of the study.

## HOMA-IR

After starting the HFFD, HOMA-IR increased from 1.3 (1.0–2.0) at baseline to 1.8 (1.2–3.4) at Week 4 ($p = 0.023$), with no significant fluctuations in median HOMA-IR values thereafter (Fig 4).

## Changes in the insulinogenic index

The insulinogenic index decreased during the first eight weeks of the HFFD, irrespective of whether it was measured over 30 or 120 minutes or whether it was indexed using the rise in insulin or in C-peptide (Fig 5A–5D). However, from Week 8 to Week 16 the insulinogenic index displayed a recovery trend when indexed using insulin (Fig 5A and 5B).

When the insulinogenic index was calculated using C-peptide, the 30-minute index remained at ≈55% of the baseline at both Week 8 and Week 16 (Fig 5C). On the other hand, when measured over 120 minutes the index improved from 60% of baseline at Week 8 to 70% of baseline by Week 16 (Fig 5D). These observations suggest that the dogs were more capable of elevating insulin during the later stages of glucose absorption than the earlier stages.

## Oral glucose minimal model parameters

The introduction of the HFFD during the initial four weeks was associated with a significant decrease in $S_I$ (from 33.1 [24.4–44.6] to 16.3 [11.2–28.1] $\cdot 10^{-4} \cdot$ dL/kg/min per μU/mL,

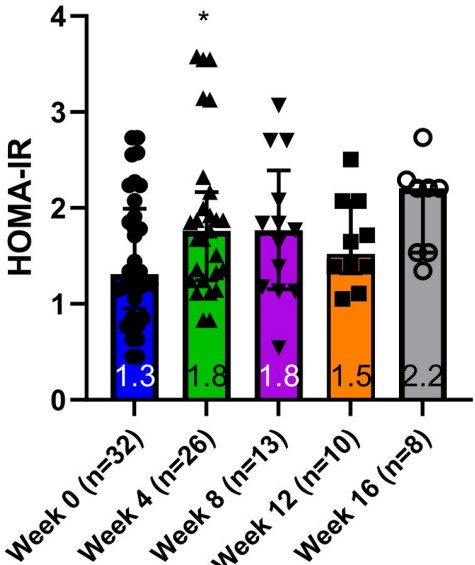

**Fig 4. Homeostatic Model Assessment of Insulin Resistance (HOMA-IR).** Measurements were made during the high-fat, high-fructose intervention, immediately prior to canine OGTT experiments. Graphs depict medians (interquartile range). * denotes $p < 0.05$ vs. Week 0.

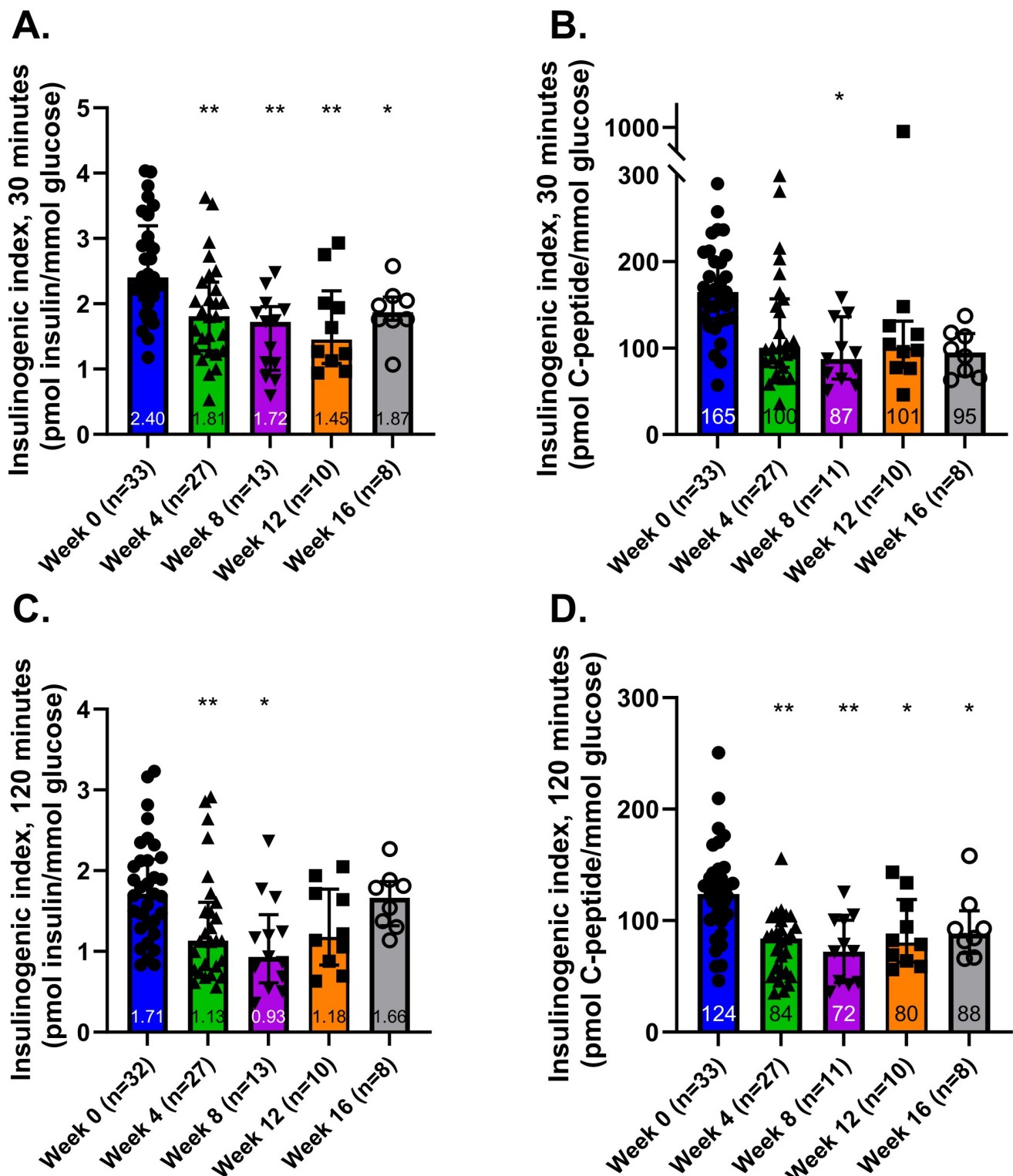

**Fig 5. Insulinogenic index measurements.** The insulinogenic index represents the ability of the body to raise insulin in response to a standard enteric glucose load. Insulinogenic indices were calculated by the ratio of: insulin AUC / glucose AUC during the first 30 minutes of the OGTT (A), C-peptide AUC / glucose AUC during the

first 30 minutes of the OGTT (B), insulin AUC / glucose AUC during the 120 minutes of the OGTT (C) and C-peptide AUC / glucose AUC during the first 120 minutes of the OGTT (D). Graphs depict medians (interquartile range). * denotes p < 0.05 and ** denotes p < 0.001.

p < 0.001), Φ (from 9.49 [7.13–14.06] to 5.13 [2.99–7.71] $\cdot 10^{-9} \cdot$ min$^{-1}$, p < 0.001) and DI (from 333.6 [159–633] to 93 [45–195] $\cdot 10^{-13}$ dL/kg min$^{-2}$ per µU/mL, p < 0.001). From Week 4 through Week 16, these parameters remained lower when compared to baseline, demonstrating a sustained impact of the HFFD on insulin sensitivity and β-cell responsivity (Fig 6).

Throughout the duration of the study, no known adverse events were observed in the dogs during the HFFD intervention.

## Discussion

This study investigated the potential of a canine model subjected to a hypercaloric HFFD to simulate the progression of human prediabetes to T2DM. Our major findings indicate that the HFFD in dogs not only caused weight gain but also induced metabolic changes analogous to early-stage human prediabetes and T2DM, including insulin resistance and β-cell dysfunction. Specifically, we observed a steep decline in insulin sensitivity, a substantial and sustained decrease in early phase insulin release, and an associated decrease in the disposition index. These findings bear striking parallels with the pathophysiology of T2DM in humans, thereby underscoring the utility of this canine model.

### The HFFD intervention was associated with a steep and sustained decrease in insulin sensitivity

Following the introduction of the HFFD, the canines exhibited a swift and enduring decrease in insulin sensitivity, with a 50% decline observed as early as four weeks into the intervention. This decrease in insulin sensitivity was maintained throughout the study, aligning with the trajectory seen in humans as glucose tolerance shifts from normal to impaired, a critical step in progression towards T2DM [19–21]. Studies employing the hyperinsulinemic, euglycemic clamp [22–24] or the intravenous glucose tolerance test [25,26] have shown that patients with IGT have 30–50% lower insulin sensitivity than those with normal glucose tolerance. Moreover, the decline in insulin sensitivity appears to just precede a decrease in early-phase insulin release in the progression of IGT [27] and T2DM [19]. This trajectory is convincingly replicated in our HFFD canine model, highlighting its potential in investigating the early onset of insulin insensitivity that characterizes the transition from IGT to T2DM in humans.

### Insulin rise relative to the glucose rise declined rapidly during the HFFD intervention and partially recovered due to an apparent reduction in hepatic insulin clearance

The insulinogenic index, calculated from both insulin and C-peptide measurements, was used to evaluate the body's ability to elevate peripheral plasma insulin levels in response to glucose increase. This index provides a measure of β-cell function, with the nature of the measurement differing depending on whether insulin or C-peptide is considered.

When using insulin to calculate the index, the result reflects not only insulin production but also insulin clearance. This is due to the fact that insulin, once produced, undergoes uptake by the liver and conversely, C-peptide, a byproduct of insulin production, is taken up by the liver in negligible amounts. Thus, the insulinogenic index computed using C-peptide presents a perspective on β-cell function that is a direct reflection of insulin production, independent of hepatic clearance.

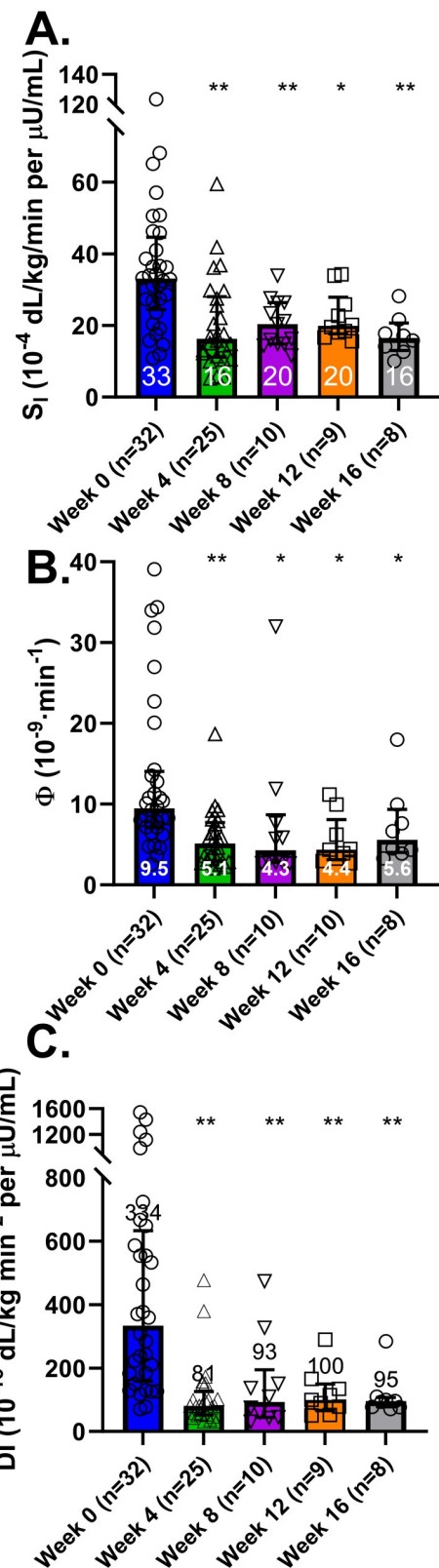

**Fig 6. Oral glucose minimal model (OGMM) data.** The OGMM was used to determine insulin sensitivity ($S_I$, A), β-cell responsivity (Φ, i.e., amount of secreted insulin per unit increase in glucose, B), and disposition index (DI, i.e., ability of the b-cell compensate for changes in insulin sensitivity, C). Graphs depict medians (interquartile range). * denotes p < 0.05 and ** denotes p < 0.001 vs. Week 0.

In the first eight weeks of the HFFD, the insulinogenic index computed using C-peptide decreased to 60% of baseline levels, during both the initial 30 and 120 minutes of the OGTT. While a modest recovery to 75% of baseline levels was observed from weeks 8 to 16 when measured over 120 minutes, no substantial recovery was noted over 30 minutes. This persistent decline suggests that the HFFD induced significant β-cell dysfunction, characterized by a sustained reduction in early-phase insulin release. The partial recovery over 120 minutes, however, indicates an attempt by the β-cell to compensate for the ongoing decrease in insulin sensitivity and early-phase insulin release.

When insulin was used to calculate the insulinogenic index, a similar trend was observed, with a recovery between weeks 8 and 16 more pronounced than calculated using C-peptide. By week 16, the 30-minute index had rebounded to 75% of baseline and the 120-minute index had rebounded to 90% of baseline levels. These patterns suggest that the body initially compensates for diminished insulin sensitivity mainly through decreasing first-pass hepatic clearance, rather than increasing insulin secretion [28].

Our assessment of β-cell responsivity, Φ, further illuminates the impact of the HFFD. The metric reflects the β-cells' ability to increase insulin secretion in response rising glucose levels. By the fourth week of the intervention, we observed a 46% drop in Φ, a decrease that remained evident throughout the HFFD intervention.

These observations draw several parallels to the course of IGT and T2DM in humans. First, the blunted early insulin response to glucose ingestion in the canine model mirrors the consistent finding in patients with IGT [21,29–31]. Second, while individuals with IGT show an enhanced late-phase insulin release compared to those with normal glucose tolerance, this enhancement is offset when adjusted for the increase in glucose levels [29]. This suggests that the compensatory mechanisms during the progression to T2DM are limited, with minimal compensation observed in the early phase, but some noticeable compensation in the late phase. Third, a recent longitudinal study in Native Americans demonstrated that a reduction in insulin clearance elevates the risk of developing T2DM, even after adjusting for other risk factors like age, body fat percentage, and early phase insulin response [32].

This evidence underlines the relevance of the HFFD canine model in emulating β-cell dysfunction observed in the progression to human T2DM, given its demonstrated diminished response to increasing glucose levels.

## HFFD intervention was associated with a rapid and sustained drop in the disposition index

The Disposition Index (DI) serves as a marker of the β-cell's ability to compensate for a decline in insulin sensitivity. In our experiments, we noted a significant downturn in insulin sensitivity by the fourth week, a trend that remained consistent throughout the duration of the study. Instead of observing a compensatory elevation in β-cell responsivity, Φ, we witnessed a decrease in this parameter as well. Consequently, by the end of Week 4, the median DI had dropped to less than one-quarter of its initial value. This observation is in line with previously reported findings, wherein an early fall in the DI precedes a rise in glucose levels, marking the progression towards T2DM [33,34]. This congruity further emphasizes the aptness of our canine model in mirroring key facets of human T2DM progression.

Some limitations of our study warrant consideration. First, while all dogs in the study received the same HFFD intervention, this is a retrospective analysis that merges data from dogs which were fed the HFFD for the purpose of testing various therapies and devices in a large-animal model of IGT and T2DM. A more rigorous approach would have been a prospective study with predefined objectives and analyses before the initiation of the experiment. Additionally, our methodology involved several assumptions due to the application of the oral glucose minimal model to canines, since the model was originally designed and validated for human use. We note that the basal C-peptide concentrations in our canine model were lower than levels typically seen in humans, while insulin is similar. Such a difference may suggest C-peptide clearance in dogs is higher than in humans. This factor could influence the direct applicability of the van Cauter formulas, which were originally developed for humans, to our canine model. Potential misalignments in C-peptide kinetics could alter the absolute values derived for $\Phi$ and DI. However, any discrepancies introduced would consistently affect all the experiments, preserving the validity of relative comparisons. Thus, while one can extrapolate our findings to human physiology, we advise caution in interpreting absolute, as opposed to relative, changes in $\Phi$ and DI.

Although this canine model provides important insights into the metabolic changes that characterize the progression to T2DM, we note that other large animal models may offer beneficial perspectives also. Dogs and pigs both have pancreas and islet structures, total β-cell masses, ratios of β-cell mass to body mass, and β-cell replication capacities that are similar to humans [10]. Hsu and colleagues showed that miniature pigs consuming a high-fat, high-fructose diet developed obesity, skeletal muscle insulin resistance, and impaired glucose tolerance [35]. Likewise, Savary-Auzeloux et al. used miniature pigs to examine how alterations in hepatic metabolite fluxes facilitated continued euglycemia during a prolonged overfeeding intervention [36]. While the use of other large animal models such as the pig may also provide valuable insights into the development of T2DM, our study underscores the efficacy of the canine model in illuminating the intricate processes underpinning human prediabetes and T2DM progression.

This study also underscores an important distinction between canine and murine models in simulating human prediabetes and T2DM. Bruce et al. recently compared the metabolic responses to OGTTs in mice and humans across varying glucose tolerance levels [7]. Their findings revealed that humans with normal and impaired glucose tolerance exhibited a substantial and prolonged insulin response, with peak concentrations exceeding 11-fold basal levels and remaining elevated for at least 150 minutes. In contrast, C57Bl/6 mice subjected to an 8-week high-fat, high-sucrose diet demonstrated a more modest insulin response, peaking at less than twofold basal levels and returning to baseline within 45 minutes post-glucose administration. Notably, the insulin response in our canine model following a HFFD intervention more closely mirrors the human response, with plasma insulin concentrations peaking at 7 to 11-fold above basal levels and remaining elevated for 120–150 minutes. This similarity underscores the potential of the canine model in providing a more accurate representation of human insulin dynamics in prediabetes and T2DM research.

Moving forward, one of the key avenues of research to explore is the reversibility of the metabolic changes induced by the HFFD. Having established the utility of the canine model in simulating human prediabetes and T2DM progression, it would be intriguing to investigate whether a return to a regular diet or implementation of dietary and therapeutic interventions could reverse the observed effects. Such a study could provide critical insights into the temporal dynamics of insulin resistance and β-cell dysfunction, and it would shed light on the potential for recovery and the key factors influencing it. Further refining the mathematical model or developing a canine-specific model could also enhance the precision of these studies, helping

to minimize assumptions and improve the interpretability of data. Moreover, our study's findings suggest that the canine model can be a valuable tool for testing novel therapeutic strategies aimed at both mitigating and reversing the progression of IGT and T2DM. These future research directions have the potential to enhance our understanding of T2DM and inform the development of innovative and effective interventions in humans.

## Conclusion

In conclusion, this study highlights the potential of using a canine model subjected to a HFFD to simulate the progression of human prediabetes and T2DM. Our data reveal that a HFFD in dogs mirrors early-stage metabolic changes in humans, such as reduced insulin sensitivity and β-cell dysfunction, providing a relevant biological model. The findings shed light on the intricate mechanisms of glucose metabolism and insulin resistance, enriching our understanding of T2DM onset and progression. This model offers an innovative and readily available avenue for the development and testing of therapeutic interventions for IGT and T2DM in the future.

## Supporting information

**S1 Dataset.**
(XLSX)

**S1 File.**
(PDF)

## Acknowledgments

The authors thank Ben Farmer, Philip Williams, Amy Nunnally, Donna Porter, Heather Sara, and Jamie Adcock for their support in caring for the animals during the research studies.

## Author Contributions

**Conceptualization:** Justin M. Gregory, Guillaume Kraft, Melanie F. Scott, Mary C. Moore, Alan D. Cherrington.

**Formal analysis:** Justin M. Gregory, Guillaume Kraft, Chiara Dalla Man, James C. Slaughter, Melanie F. Scott, Dale S. Edgerton, Mary C. Moore, Alan D. Cherrington.

**Funding acquisition:** Alan D. Cherrington.

**Investigation:** Justin M. Gregory, Guillaume Kraft, Melanie F. Scott, Jon R. Hastings, Mary C. Moore, Alan D. Cherrington.

**Methodology:** Justin M. Gregory, Guillaume Kraft, Chiara Dalla Man, James C. Slaughter, Melanie F. Scott, Dale S. Edgerton, Mary C. Moore, Alan D. Cherrington.

**Project administration:** Jon R. Hastings, Alan D. Cherrington.

**Software:** Chiara Dalla Man, James C. Slaughter.

**Supervision:** Alan D. Cherrington.

**Visualization:** Justin M. Gregory.

**Writing – original draft:** Justin M. Gregory.

**Writing – review & editing:** Guillaume Kraft, Chiara Dalla Man, James C. Slaughter, Melanie F. Scott, Jon R. Hastings, Dale S. Edgerton, Alan D. Cherrington.

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
