## [Decision Letter · Decision Letter 0]

7 Nov 2023

PONE-D-23-29774A High-Fat and Fructose Diet in Dogs Mirrors Insulin Resistance and β-cell Dysfunction Characteristic of Impaired Glucose Tolerance in HumansPLOS ONE

Dear Dr. Gregory,

Thank you for submitting your manuscript to PLOS ONE. After careful consideration, we feel that it has merit but does not fully meet PLOS ONE’s publication criteria as it currently stands. Therefore, we invite you to submit a revised version of the manuscript that addresses the points raised during the review process.

When you're ready to submit your revision by Dec 22 2023 11:59PM, log on to https://www.editorialmanager.com/pone/ and select the 'Submissions Needing Revision' folder to locate your manuscript file.

Please include the following items when submitting your revised manuscript:A rebuttal letter that responds to each point raised by the academic editor and reviewer(s). You should upload this letter as a separate file labeled 'Response to Reviewers'.A marked-up copy of your manuscript that highlights changes made to the original version. You should upload this as a separate file labeled 'Revised Manuscript with Track Changes'.An unmarked version of your revised paper without tracked changes. You should upload this as a separate file labeled 'Manuscript'.If applicable, we recommend that you deposit your laboratory protocols in protocols.io to enhance the reproducibility of your results. Protocols.io assigns your protocol its own identifier (DOI) so that it can be cited independently in the future. For instructions see: https://journals.plos.org/plosone/s/submission-guidelines#loc-laboratory-protocols. Additionally, PLOS ONE offers an option for publishing peer-reviewed Lab Protocol articles, which describe protocols hosted on protocols.io. Read more information on sharing protocols at https://plos.org/protocols?utm_medium=editorial-email&utm_source=authorletters&utm_campaign=protocols.

We look forward to receiving your revised manuscript.

Kind regards,

Chikezie Hart Onwukwe

Academic Editor

PLOS ONE

2. 1. As part of your revision, please complete and submit a copy of the Full ARRIVE 2.0 Guidelines checklist, a document that aims to improve experimental reporting and reproducibility of animal studies for purposes of post-publication data analysis and reproducibility: https://arriveguidelines.org/sites/arrive/files/documents/Author%20Checklist%20-%20Full.pdf Please include your completed checklist as a Supporting Information file. Note that if your paper is accepted for publication, this checklist will be published as part of your article.

Reviewers' comments:

Reviewer's Responses to Questions

**Comments to the Author**

1. Is the manuscript technically sound, and do the data support the conclusions?

Reviewer #1: Yes

Reviewer #2: Yes

2. Has the statistical analysis been performed appropriately and rigorously? 

Reviewer #1: I Don't Know

Reviewer #2: Yes

3. Have the authors made all data underlying the findings in their manuscript fully available?

Reviewer #1: No

Reviewer #2: Yes

4. Is the manuscript presented in an intelligible fashion and written in standard English?

Reviewer #1: Yes

Reviewer #2: Yes

5. Review Comments to the Author

Reviewer #1: Peer Review Report for PONE-D-23-29774

General Impression

This is an interesting paper which is timely, important to the field, and highlights a section of the field that has not been explored much before, highlighting the need of such research. The manuscript is well-written and is easy to understand. While the idea might not be entirely novel, the manuscript builds on previous research, which is scarce.

Introduction

The introduction is of a good length, relevant to the topic, and does not go beyond the topic to include unnecessary information. I have a few minor comments that the authors can consider.

[1] Page 4, First Paragraph:

The authors say “... pose an increasingly significant global health burden.”. Consider quoting the burden quantitatively through prevalence statistics, or similar, to add weight to the statement.

[2] Page 4, Second Paragraph:

The authors say “Murine models, such as the leptin receptor deficient db/db mouse, are a popular choice for… given their genetic tractability and cost-effectiveness”. The authors can consider mentioning the time-saving benefit of using db/db mice, as well as the removed need for invasive interventions.

[3] Page 4, Second Paragraph:

The authors say “Murine models, such as the leptin receptor deficient db/db mouse, are a popular choice for… given their genetic tractability and cost-effectiveness”. Consider citing a more recent reference in addition to the study cited already (Hummel KP, 1966). Authors could consider highlighting how murine models have served research purposes in the relevant field for so long.

[4] Page 4, Second Paragraph:

The authors say “Despite these models' ability to emulate some aspects of human T2DM such as obesity,…”. Consider revising the statement to make it more specific as genetic obesity in db/db mice is achieved through mechanisms different to the ones in T2DM-relevant obesity. Authors could consider moving this part of the sentence somewhere else, or restructure the sentence to make it more accurate.

[5] Page 4, Last Paragraph:

The authors say “With a level of genetic diversity similar to that of humans, canine models…”, but do not quote a reference. Consider citing a recent reference to support the statement.

Methods

I found this section detailed enough to make this research paper reproducible. The structure and the language used make the section easy to comprehend for anyone who is new to the field. However, I feel that some essential information is lacking and therefore I have a few comments for this section.

[1] Did the authors use a checklist (e.g. ARRIVE) to ensure quality of the manuscript? If so, the authors should consider saying so.

[2] Is the protocol for this study available? If so, the authors should consider citing or mentioning where it can be accessed.

[3] To assist the readers of the manuscript, the authors can consider incorporating a visual figure to highlight the timeline and interventions.

[4] The authors can consider explicitly stating that no control group was present, and state a rationale for the decision.

[5] The authors should also consider stating briefly where the animals, which contributed to the study as subjects, were kept, their immunization status, and any other relevant information to highlight the general condition of their habitat.

[6] Do the authors have a justification for the sample size. If so, they may consider stating the rationale behind it.

[7] Did the authors blind the statisticians to relevant data to ensure no bias was present? If so, or if not, the authors should consider stating their decision and the rationale behind it.

Results

I have no comment for this section as the authors have ensured a good structural approach to highlight all relevant information.

Discussion

The discussion does justice to the findings of this study. However, I have two comments.

[1] Page 17, Last Paragraph

The authors say “…this is a retrospective analysis that merges data from dogs which were fed the HFFD for the purpose of testing various…”. However, the authors make no explicit mention of this fact in the Method section. The authors should consider stating the appropriate statement in the appropriate section as well.

[2] Currently, the authors discuss the relevance of the canine model to humans. Therefore, the discussion would benefit if the authors also discuss the results of their study by offering a comparison with existing/current animal models. This is especially important in light of the manuscript itself, where the authors begin the manuscript by highlighting the deficiency murine models have, and extrapolate this deficiency, in part, to build the rationale behind this study.

Figures

[1] The cluster of plots in Figure 1 (and other bar charts) is too crowded to properly read. The authors can consider highlighting the change in weight of each animal in a table in a supplement to make the figure easier to read and make data available to the audience.

Reviewer #2: The article presents a novel approach for animal models , the methodology is well written , results were adequately presented and discussed

Conclusions are well in accordance with presented results with study limitations mentioned

6. PLOS authors have the option to publish the peer review history of their article (what does this mean?). If published, this will include your full peer review and any attached files.

Reviewer #1: No

Reviewer #2: **Yes: **Yara Muhammed Eid

---

## [Author Response · Author response to Decision Letter 0]

1 Dec 2023

November 13, 2023

Chikezie Hart Onwukwe, M.D.

Division of Endocrinology and Diabetes, Al Isawiya General Hospital

Ministry of Health, Directorate of Gurayat

Kingdom of Saudi Arabia

Dear Dr. Onwukwe,

I sincerely appreciate the reviewers’ thoughtful critiques of our manuscript, entitled, “A High-Fat and Fructose Diet in Dogs Mirrors Insulin Resistance and β-cell Dysfunction Characteristic of Impaired Glucose Tolerance in Humans.” Their suggestions have improved the quality of our paper, and we are hopeful that you and the reviewers will receive our revisions favorably. 

In the response that follows, we respond to each point raised by you as the academic editor and the reviewers. Our responses to the reviewers are written in indented, italicized paragraphs. We have uploaded a marked-up copy of the manuscript that highlights changes made to the original version in red. We also upload an unmarked version of the revised paper without tracked changes.

Thank you again for the time you spend as an academic editor at PLOS One. I hope this letter finds you well.

Yours, respectfully and sincerely,

Justin M. Gregory, M.D., M.S.C.I.

Assistant Professor of Pediatrics

Division of Pediatric Endocrinology, Vanderbilt University School of Medicine

Nashville, Tennessee, USA

 

Responses to Academic Editor: 

1. Ensuring PLOS ONE’s style requirements: I have re-read PLOS One’s style requirements to align our manuscript to the style templates. I have altered the manuscript’s title and section headings to sentence case throughout. 

2. Submitting a copy of the Full ARRIVE 2.0 guidelines checklist: I have completed the checklist and made mostly minor revisions to the manuscript to ensure the items in the checklist were satisfied. This checklist is now included as a supporting information file.

3. Additional information regarding animals: I have added a paragraph onto the end of the materials and methods section, experimental design subsection. This paragraph describes methods to enrich the environment as appropriate to the species, consistent with U.S. Public Health Service Guide to the Care and Use of Laboratory Animals. We also describe the use of sodium pentobarbital as the method of euthanasia. Because the only invasive procedure required in the OGTT experiments included placement of an intravenous angiocatheter, which does not require anesthesia and/or analgesia, we did not specifically discuss methods of anesthesia or analgesia. 

4. Including raw data: In this revision, we now include raw data as a supplementary datafile. 

5. ‘Funding Information’ and ‘Financial Disclosure’ sections do not match: Under funding information in the online submission system at editorialmanager.com, we have tried to include Fractyl Health and Metavention as two companies that supported the research. I have been unable to add these companies in the online submission system, however. 

I have also been unable to find a section in the online submission system to align the Funding Information with the Financial Disclosure section. In the revised manuscript word document, on page. 26 of the “tracked changes version,” I have inserted a section entitled “Funding Information,” which specifies the role of the fundings sources. 

Additionally, I have changed the section title in the manuscript from “Conflicts of Interest” to “Competing Interests” to more properly align with PLOS ONE’s author instructions. 

If additional revisions are needed, I am happy to make any further modifications in this regard. 

6. Minimal data set: In the final sentence of the revised methods section, we now indicate that that study data are included in a supplemental dataset. 

7. References: We have reviewed the reference list to ensure it is complete and correct. 

 

Responses to Reviewers: 

Reviewers' comments:

Reviewer's Responses to Questions 

Comments to the Author

1. Is the manuscript technically sound, and do the data support the conclusions?

Reviewer #1: Yes

Reviewer #2: Yes

We appreciate the reviewers’ comments that the manuscript is technically sound, and that the data support the conclusions.

2. Has the statistical analysis been performed appropriately and rigorously? 

Reviewer #1: I Don't Know

Reviewer #2: Yes

We appreciate reviewer 1’s candor and reviewer 2 agreeing that the statistical analysis has been performed appropriately and rigorously. With the revised submission, we now include the raw data for review. 

3. Have the authors made all data underlying the findings in their manuscript fully available?

Reviewer #1: No

Reviewer #2: Yes

As noted above, with the revised submission we now include the raw data for review. 

4. Is the manuscript presented in an intelligible fashion and written in standard English?

Reviewer #1: Yes

Reviewer #2: Yes

We appreciate the reviewers’ comments that the manuscript is presented in an intelligible fashion and written in standard English. 

5. Review Comments to the Author

Reviewer #1: Peer Review Report for PONE-D-23-29774

General Impression

This is an interesting paper which is timely, important to the field, and highlights a section of the field that has not been explored much before, highlighting the need of such research. The manuscript is well-written and is easy to understand. While the idea might not be entirely novel, the manuscript builds on previous research, which is scarce.

We appreciate Reviewer 1’s positive general impressions regarding the manuscript and the comments that follow. 

Introduction

The introduction is of a good length, relevant to the topic, and does not go beyond the topic to include unnecessary information. I have a few minor comments that the authors can consider.

[1] Page 4, First Paragraph:

The authors say “... pose an increasingly significant global health burden.”. Consider quoting the burden quantitatively through prevalence statistics, or similar, to add weight to the statement.

We have revised this sentence with more specific prevalence statistics drawn from the IDF Diabetes Atlas. 

[2] Page 4, Second Paragraph:

The authors say “Murine models, such as the leptin receptor deficient db/db mouse, are a popular choice for… given their genetic tractability and cost-effectiveness”. The authors can consider mentioning the time-saving benefit of using db/db mice, as well as the removed need for invasive interventions.

We have modified this sentence to emphasize how the time saving benefit of this model contributes to cost-effective research. 

[3] Page 4, Second Paragraph:

The authors say “Murine models, such as the leptin receptor deficient db/db mouse, are a popular choice for… given their genetic tractability and cost-effectiveness”. Consider citing a more recent reference in addition to the study cited already (Hummel KP, 1966). Authors could consider highlighting how murine models have served research purposes in the relevant field for so long.

We have further modified this sentence to make clear the model is well established and now cite a 2022 article by Ho Lee, who reviews how studies using models including ob/ob and db/db mice have revealed pathophysiological links between insulin resistant states and predisposal to obesity-associated cancers. 

[4] Page 4, Second Paragraph:

The authors say “Despite these models' ability to emulate some aspects of human T2DM such as obesity,…”. Consider revising the statement to make it more specific as genetic obesity in db/db mice is achieved through mechanisms different to the ones in T2DM-relevant obesity. Authors could consider moving this part of the sentence somewhere else, or restructure the sentence to make it more accurate.

We have revised this statement substantially to reflect the complexities of obesity in human T2DM.

[5] Page 4, Last Paragraph:

The authors say “With a level of genetic diversity similar to that of humans, canine models…”, but do not quote a reference. Consider citing a recent reference to support the statement.

In the revised manuscript we now cite an extensive review of the canine genome by Ostrander and Wayne. The authors note that humans and dogs have similar levels of overall nucleotide diversity and that the patterns of genetic linkage disequilibrium are analogous to human populations. They also note that hundreds of genetics disorders found in humans have also been described in dogs. 

Methods

I found this section detailed enough to make this research paper reproducible. The structure and the language used make the section easy to comprehend for anyone who is new to the field. However, I feel that some essential information is lacking and therefore I have a few comments for this section.

[1] Did the authors use a checklist (e.g. ARRIVE) to ensure quality of the manuscript? If so, the authors should consider saying so.

In the last sentence of the methods section in our revised manuscript, we now make explicit that the ARRIVE 2.0 checklist is included as supplemental information. 

[2] Is the protocol for this study available? If so, the authors should consider citing or mentioning where it can be accessed.

With the resubmission, we now include a copy of a protocol used in the research. 

[3] To assist the readers of the manuscript, the authors can consider incorporating a visual figure to highlight the timeline and interventions.

We now include a timeline figure in the methods section. 

[4] The authors can consider explicitly stating that no control group was present, and state a rationale for the decision.

We now make this explicit in the final sentence of the “animals and experimental timeline” paragraph of the methods section. 

[5] The authors should also consider stating briefly where the animals, which contributed to the study as subjects, were kept, their immunization status, and any other relevant information to highlight the general condition of their habitat.

To provide more details regarding these conditions, we have provided several additional details throughout the first paragraph of the “animals and experimental timeline.” We have also added a paragraph to the “experimental design” section discussing elaborating on the methods used to enrich the environment appropriate to the canine species. 

[6] Do the authors have a justification for the sample size. If so, they may consider stating the rationale behind it.

In the revision, we now discuss the lack of an a priori sample size calculation in the final sentence of the “animals and experimental timeline” subsection. 

[7] Did the authors blind the statisticians to relevant data to ensure no bias was present? If so, or if not, the authors should consider stating their decision and the rationale behind it.

Because of the retrospective and single-arm nature of the high-fat, high-fructose intervention (HFFD) no blinding procedure was used for investigators conducting the statistical analysis of the data. In the revised manuscript, we make this explicit in the final two sentences of the “animals and experimental timeline” subsection. 

Results

I have no comment for this section as the authors have ensured a good structural approach to highlight all relevant information.

We appreciate Reviewer 1’s positive feedback in this regard. 

Discussion

The discussion does justice to the findings of this study. However, I have two comments.

[1] Page 17, Last Paragraph

The authors say “…this is a retrospective analysis that merges data from dogs which were fed the HFFD for the purpose of testing various…”. However, the authors make no explicit mention of this fact in the Method section. The authors should consider stating the appropriate statement in the appropriate section as well.

We now mention that the dogs completed their series of OGTTs, they subsequently underwent preclinical testing of differing diabetes and devices in the new final paragraph of the “experimental design” subsection of the methods section. 

[2] Currently, the authors discuss the relevance of the canine model to humans. Therefore, the discussion would benefit if the authors also discuss the results of their study by offering a comparison with existing/current animal models. This is especially important in light of the manuscript itself, where the authors begin the manuscript by highlighting the deficiency murine models have, and extrapolate this deficiency, in part, to build the rationale behind this study.

We appreciate Reviewer 1’s suggestion in this regard. In the introduction, we discuss that unlike humans, mice display only a transient insulin response during OGTTs. In the revised discussion we have now added an additional paragraph circling back around to this point. We discuss how the insulin response to OGTTs in dogs more closely resembles humans than OGTTs in mice. This paragraph follows our discussion of how other large animal models may also offer insights into the metabolic changes that characterize the progression to T2DM. 

Figures

[1] The cluster of plots in Figure 1 (and other bar charts) is too crowded to properly read. The authors can consider highlighting the change in weight of each animal in a table in a supplement to make the figure easier to read and make data available to the audience.

We thank Reviewer 1 for this suggestion. Our resubmission includes a dataset from our study with one spreadsheet listing each animal’s weight. 

Reviewer #2: The article presents a novel approach for animal models , the methodology is well written , results were adequately presented and discussed

Conclusions are well in accordance with presented results with study limitations mentioned

We are grateful for Reviewer 2’s favorable review of our manuscript and appreciate his or her time and consideration.

---

## [Decision Letter · Decision Letter 1]

13 Dec 2023

A High-Fat and Fructose Diet in Dogs Mirrors Insulin Resistance and β-cell Dysfunction Characteristic of Impaired Glucose Tolerance in Humans

PONE-D-23-29774R1

Dear Dr. Gregory,

We’re pleased to inform you that your manuscript has been judged scientifically suitable for publication and will be formally accepted for publication once it meets all outstanding technical requirements.

Kind regards,

Chikezie Hart Onwukwe

Academic Editor

PLOS ONE

Additional Editor Comments (optional):

Reviewers' comments:

Reviewer's Responses to Questions

**Comments to the Author**

1. If the authors have adequately addressed your comments raised in a previous round of review and you feel that this manuscript is now acceptable for publication, you may indicate that here to bypass the “Comments to the Author” section, enter your conflict of interest statement in the “Confidential to Editor” section, and submit your "Accept" recommendation.

Reviewer #1: All comments have been addressed

2. Is the manuscript technically sound, and do the data support the conclusions?

Reviewer #1: Yes

3. Has the statistical analysis been performed appropriately and rigorously? 

Reviewer #1: I Don't Know

4. Have the authors made all data underlying the findings in their manuscript fully available?

Reviewer #1: Yes

5. Is the manuscript presented in an intelligible fashion and written in standard English?

Reviewer #1: Yes

6. Review Comments to the Author

Reviewer #1: I have gone through the entire manuscript, including the additional files that were submitted as part of the revised submission. I maintain my earlier impression of the manuscript by stating that this research study is timely and important to the field.

All my prior comments have been addressed and I have not seen the need to raise any further questions regarding the quality, nor the validity, of this manuscript.

I thank the authors for their work.

Warm Regards

7. PLOS authors have the option to publish the peer review history of their article (what does this mean?). If published, this will include your full peer review and any attached files.

Reviewer #1: No

---

## [Editor Report · Acceptance letter]

14 Dec 2023

PONE-D-23-29774R1 

PLOS ONE

Dear Dr. Gregory, 

I'm pleased to inform you that your manuscript has been deemed suitable for publication in PLOS ONE. Congratulations! Your manuscript is now being handed over to our production team.

Kind regards, 

on behalf of

Dr. Chikezie Hart Onwukwe 

Academic Editor

PLOS ONE